# Anti-Melanogenic Effects of *Lilium lancifolium* Root Extract via Downregulation of PKA/CREB and MAPK/CREB Signaling Pathways in B16F10 Cells

**DOI:** 10.3390/plants12213666

**Published:** 2023-10-24

**Authors:** Seokmuk Park, Nayeon Han, Jungmin Lee, Jae-Nam Lee, Sungkwan An, Seunghee Bae

**Affiliations:** 1Department of Cosmetics Engineering, Konkuk University, 120 Neungdong-ro, Gwangjin-gu, Seoul 05029, Republic of Korea; ted968@konkuk.ac.kr (S.P.); nina1305@konkuk.ac.kr (N.H.); 2Dermato Bio, Inc., #505, Techno Cube, 13-18 Songdogwahak-ro 16beon-gil, Yeongsu-gu, Incheon 21984, Republic of Korea; jungmin-lee@skinbutak.com; 3Department of Cosmetology, Graduate School of Engineering, Konkuk University, 120 Neungdong-ro, Gwangjin-gu, Seoul 05029, Republic of Korea; jn386@konkuk.ac.kr; 4Eco Up Bio, Inc., 373 Chang-ui-ri, Seorak-myeon, Gapyeong-gun 477852, Republic of Korea; anxcompany@naver.com

**Keywords:** melanogenesis, anti-melanogenic effect, melanin, B16F10, α-melanocyte stimulating hormone (α-MSH), *Lilium lancifolium*, L-phenylalanine, regaloside A

## Abstract

Hyperpigmentation disorders causing emotional distress require the topical use of depigmenting agents of natural origin. In this study, the anti-melanogenic effects of the *Lilium lancifolium* root extract (LRE) were investigated in B16F10 cells. Consequently, a non-cytotoxic concentration of the extract reduced intracellular melanin content and tyrosinase activity in a dose-dependent manner, correlating with the diminished expression of core melanogenic enzymes within cells. LRE treatment also inhibited cyclic adenosine monophosphate (cAMP) response element-binding protein (CREB)/microphthalmia-associated transcription factor signaling, which regulates the expression of tyrosinase-related genes. Upon examining these findings from a molecular mechanism perspective, LRE treatment suppressed the phosphorylation of protein kinase A (PKA), p38, and extracellular signal-related kinase (ERK), which are upstream regulators of CREB. In addition, L-phenylalanine and regaloside A, specifically identified within the LRE using liquid chromatography-mass spectrometry, exhibited inhibitory effects on melanin production. Collectively, these results imply that LRE potentially suppresses cAMP-mediated melanogenesis by downregulating PKA/CREB and mitogen-activated protein kinase (MAPK)/CREB signaling pathways. Therefore, it can be employed as a novel therapeutic ingredient of natural origin to ameliorate hyperpigmentation disorders.

## 1. Introduction

Melanin is synthesized within specialized ovoid organelles called melanosomes, which are produced by dendritic melanocytes in the basal layer of the epidermis, constituting only 1% of the total composition [1]. Melanin produced within melanosomes is transported via dendrites to neighboring keratinocytes, where it accumulates in the perinuclear area of both keratinocytes and melanocytes and forms supranuclear “caps” recognized for their ability to shield DNA from ultraviolet (UV) rays [2]. In addition, melanin shields skin cells from the harmful effects of UV radiation, oxidative stress, and other environmental factors [3,4,5]. Eumelanin is a deep brown/black insoluble polymer, whereas pheomelanin is a light red/yellow soluble polymer containing sulfur [1,6,7,8]. Eumelanin plays a crucial role in protecting against and determining the color of the skin, hair, and eyes. Excessive melanin expression has been attributed to conditions, such as melanoma, freckles, lentigo, and abnormal skin pigmentation [9,10,11,12,13]. Among these disorders, hyperpigmentation, which is characterized by the excessive production of melanin, frequently originates from increased melanocyte and tyrosinase activity [14,15]. Previous studies have explored hyperpigmentation-related diseases, such as melasma, post-inflammatory hyperpigmentation (PIH), and lentigines, each with distinct characteristics and potential side effects [16]. Melasma is a prevalent hyperpigmentation disorder that is frequently triggered by hormonal changes [15]. It can lead to emotional distress, diminished self-esteem, and feelings of frustration due to the noticeable visibility of the condition [17,18]. Addison’s disease is a hormonal disorder that can cause increased melanin production. Side effects of Addison’s disease include mental health disorders and abnormal menstrual cycles [15].

The α-melanocyte-stimulating hormone (α-MSH) is primarily used in cultured melanocytes as a cell model in vivo [19]. DNA damage and repair are associated with the stimulation of melanogenesis, supported by p53 activation, which leads to elevated levels of tyrosinase messenger RNA (mRNA) and protein [20]. Notably, p53 activation triggers transcription from the proopiomelanocortin (POMC) gene promoter, inducing the release of α-MSH from keratinocytes, a pivotal inducer of melanogenesis [21]. α-MSH then stimulates the melanocortin-1 receptor (MC1R) in melanocytes, promoting eumelanin production [21]. Therefore, melanin reduction induced by α-MSH can be a major strategy for pigment treatment.

Hydroquinone, kojic acid, ascorbic acid, and arbutin are hypopigmentation agents known for their potential to inhibit melanin production and are commonly used for skin whitening or depigmentation purposes [22,23]. However, these conventional pigment inhibitors have notable drawbacks, including skin irritation, high allergic reactivity, and adverse responses [24,25,26,27]. Hydroquinone is currently the benchmark standard topical drug for the treatment of hyperpigmentation disorders, such as melanosis [28]. However, permanent depigmentation and exogenous ochronosis have been reported with prolonged use [29,30,31,32]. Its utilization is not legally available by prescription or OTC in the European Union, Australia, or Japan owing to its unknown safety profile [29,32,33]. Arbutin, a naturally occurring beta-D-glucopyranoside found in bearberry, cranberry, and blueberry leaves, is among the foremost effective over-the-counter (OTC) pigment-lightening ingredients, with a structure similar to that of hydroquinone [34,35,36,37]. It functions by decreasing tyrosinase activity without affecting messenger RNA expression or inhibiting melanosome maturation; however, caution is warranted as it may lead to paradoxical pigment darkening due to PIH [34,37]. Kojic acid, a hydrophilic fungal derivative obtained from *Aspergillus* and *Penicillium* species, is the second most effective OTC melanogenic agent [34,38]. It is attributed to the inhibition of tyrosinase activity upon binding to copper [39,40]. This interference with tyrosinase, a pivotal enzyme in melanin production, contributes to its anti-melanogenic effects [41]. Kojic acid is a widely recognized therapy for treating melasma [42]. However, specific issues have been raised as kojic acid is recognized as a sensitizer and has shown mutagenic properties in cell culture studies [43]. Considering the potential risks associated with these inhibitors, researchers have explored novel candidates that can effectively inhibit tyrosinase without causing adverse effects [44,45]. In this endeavor, plant biosynthesized metabolites have emerged as promising alternatives to synthetic analogs [46,47]. These natural compounds exhibit potential for the development of less hazardous and more effective solutions for skin lightening and hyperpigmentation treatment.

*Lilium lancifolium*, commonly known as the tiger lily, belongs to the Liliaceae family [48,49,50]. These plants are distributed and cultivated in various temperate regions, including Eastern Asia, Europe, and North America [48,51]. In China, *L. lancifolium* is known for its edible bulbs, which possess high nutritional and antioxidative properties [52]. Furthermore, traditional Korean medicine has recognized the potential of the *Lilium* species to treat inflammatory disorders [53]. The root of *L. lancifolium* Thunb has historically been used for various respiratory conditions [49,54]. Polysaccharides from *L. lancifolium* have various beneficial effects, such as diminishing nitric oxide production in macrophages [51,55,56]. The leaves, roots, and bulbs of *L. lancifolium* contain amino acids, polysaccharides, saponins, phenylpropanoids, phenolics, and other constituents [57,58,59,60]. This composition suggests the potential capacity to improve skin conditions. However, the anti-melanogenic effects of *L. lancifolium* root extract (LRE) and the underlying molecular mechanisms have not yet been explored. Therefore, this study aimed to elucidate the potential of LRE as an agent for inhibiting hyperpigmentation and as a raw material for formulating whitening functional products in the pharmaceutical and cosmetic sectors.

## 2. Results

### 2.1. Effects of LRE on Cell Viability and Melanogenesis in B16F10 Cells

*Lilium* species contain numerous bioactive compounds, such as amino acids, polysaccharides, phenolics, and flavonoids [61]. Based on the abundance of these bioactive substances, cosmetic ingredients derived from *Lilium* plants have been utilized for anti-aging, radiation protection, moisturization, acne treatment, and hair growth promotion. However, no study has investigated the anti-melanogenic effects of *L. lancifolium* [62]. Prior to assessing the anti-melanogenic effects of LRE, its potential cytotoxicity was examined in B16F10 cells (Figure 1A). Cells were treated with various concentrations (0, 20, 50, 100, and 200 µg/mL) of LRE and α-MSH (200 nM) for 48 h. No cytotoxic effect on B16F10 cells treated with doses below 100 µg/mL of LRE was observed (Figure 1A). However, B16F10 cells treated with 200 µg/mL of LRE exhibited a 17.25% reduction in cell viability compared to the untreated group. Moreover, co-treatment of LRE (200 µg/mL) and α-MSH (200 nM) resulted in an 18.59% reduction in cell viability compared to the α-MSH-treated group. Consequently, for subsequent experiments, LRE was utilized at concentrations below 100 µg/mL.

Next, the effects of LRE on melanogenesis were evaluated. LRE significantly suppressed melanogenesis in B16F10 cells in a dose-dependent manner (Figure 1B). In the group treated with 100 µg/mL of LRE, there was a 17.76% reduction in the production of melanin compared to the untreated group. In a similar manner, co-treatment of LRE (100 µg/mL) and α-MSH (200 nM) demonstrated a 65.67% reduction in melanin content compared to the α-MSH-treated group. Arbutin (100 µM) was employed as the positive control group, and cells treated with LRE (100 µg/mL) exhibited a 19.43% reduction in melanin content compared to those treated with arbutin. Similarly, co-treatment of LRE (100 µg/mL) and α-MSH (200 nM) showed an 18.31% reduction in cellular melanin content compared to the group co-treated with α-MSH (200 nM) and arbutin (100 µM). These results suggest that the bioactive components of LRE may suppress melanin synthesis by inhibiting melanogenesis-related gene and protein expression.

### 2.2. Inhibitory Effects of LRE on the Expression of Melanogenic Enzymes

Tyrosinase, tyrosinase-related protein 1 (Tyrp1), and tyrosinase-related protein 2 (Tyrp2), collectively referred to as melanogenic enzymes, play crucial roles in pigmentation [63]. Tyrosinase catalyzes the conversion of tyrosine to 3,4-dihydroxyphenylalanine, which is a vital and rate-limiting step in melanin synthesis [64]. Similarly, Tyrp1 and Tyrp2, two other tyrosinase-related proteins, also contribute to eumelanin synthesis [65]. As LRE inhibited melanin production in B16F10 cells, we investigated whether LRE regulates the expression of melanogenic enzymes in these cells. LRE markedly suppressed cellular tyrosinase activity (Figure 2A). Cells treated with 100 µg/mL of LRE displayed a 36.33% reduction in cellular tyrosinase activity compared to the untreated group. Stimulation with α-MSH significantly increased tyrosinase enzyme function within the cells. Conversely, the co-treatment group with α-MSH (200 nM) and LRE (0–100 µg/mL) demonstrated a dose-dependent decrease in intracellular tyrosinase activity (Figure 2A). In the group co-treated with α-MSH and 100 µg/mL of LRE, an 81.91% reduction in tyrosinase activity was observed compared to the α-MSH-treated group. Similar to Figure 1B, LRE exerted a substantial inhibitory effect on cellular tyrosinase activity compared to the positive control, arbutin (100 µM). The LRE-induced reduction in tyrosinase activity can be attributed to a reduction in the expression and stability of melanogenic enzymes. To verify whether LRE reduced the expression of tyrosinase, Tyrp1, and Tyrp2 in B16F10 cells, Western blotting, reverse-transcription polymerase chain reaction (RT-PCR), and quantitative real-time polymerase chain reaction (qRT-PCR) assays were conducted following co-treatment with the specified concentrations of LRE and α-MSH. Our results revealed that LRE suppressed the mRNA expression of tyrosinase, Tyrp1, and Tyrp2 in B16F10 cells (Figure 2B and Appendix A). In particular, RT-PCR and qRT-PCR assays showed that the co-treatment group with α-MSH (200 nM) and LRE (50 and 100 μg/mL) exhibited a dose-dependent decrease in tyrosinase-related genes (Figure 2B and Appendix A). This effect was consistent with the inhibition of protein levels of tyrosinase, Tyrp1, and Tyrp2 (Figure 2C). Collectively, these findings suggest that the inhibition of melanogenesis by LRE can be attributed to the suppression of melanogenic enzyme activity and the downregulation of tyrosinase, Tyrp1, and Tyrp2 expression in B16F10 cells.

### 2.3. Inhibitory Effects of LRE on Cyclic Adenosine Monophosphate (cAMP) Response Element-Binding Protein (CREB)/Microphthalmia-Associated Transcription Factor (Mitf) Signaling Pathway

Recent studies have revealed the significance of Mitf expression in regulating melanogenic enzyme levels and influencing melanogenesis in B16F10 cells [66,67,68]. In addition, tyrosinase, Tyrp1, and Tyrp2 are the major targets of melanogenic enzymes induced by Mitf [69]. Under physiological conditions, the elevation in cAMP induced by α-MSH results in phosphorylation at the serine 133 residue of CREB via various signaling pathways [70,71,72]. This ultimately results in an increase in the transcriptional level of Mitf [73]. Therefore, we investigated whether LRE, which inhibits the expression of tyrosinase-related proteins, downregulated the CREB/Mitf signaling pathway. LRE decreased both Mitf mRNA and protein levels (Figure 3A,B and Appendix A). In particular, the elevated protein and mRNA levels of Mitf induced by α-MSH treatment were significantly reduced to levels comparable to those in the control group following LRE treatment. The phosphorylation level at the serine 133 residue of CREB, which is the most prominent transcription factor regulating the transcriptional level of Mitf, was assessed via Western blotting. α-MSH (200 nM) treatment increased the phosphorylation levels of CREB compared to the untreated control cells; however, the cells co-treated with LRE significantly decreased the phosphorylation of CREB in a dose-dependent manner compared to the α-MSH-treated cells (Figure 3C,D). Treatment with LRE (100 µg/mL) also exhibited a time-dependent reduction in α-MSH-induced Mitf and p-CREB protein expression (Figure 4A–C). These findings imply that LRE downregulates the expression of melanogenic enzymes via the CREB/Mitf signaling pathway.

### 2.4. Inhibitory Effects of LRE on Protein Kinase A (PKA)/CREB and Mitogen-Activated Protein Kinase/CREB Signaling Pathways

Several kinases target CREB, a transcription factor that regulates the transcription of Mitf [73]. These regulators promote CREB phosphorylation at the serine 133 residue, activating CREB-dependent transcription [74]. Notably, the major kinases facilitating CREB phosphorylation and its subsequent binding to Mitf promoters are PKA, p38, and extracellular signal-related kinase (ERK) [74,75,76]. In particular, the α-MSH/MC1R pathway, the initial signaling pathway inducing melanogenesis in melanocytes, elevates intracellular cAMP levels [77]. Consequently, cAMP induces the phosphorylation of PKA, p38, and ERK, resulting in the transcriptional activation of CREB [78,79,80,81]. To comprehend the molecular mechanism by which LRE inhibits the phosphorylation of CREB at the serine 133 residue, the upstream kinases of CREB were assessed via Western blotting. LRE significantly reduced the phosphorylation of PKA, p38, and ERK in a dose-dependent manner (Figure 5A,B). In particular, the group treated with 100 µg/mL of LRE showed a significant decrease in the phosphorylation of PKA, p38, and ERK compared to the untreated control group. Treatment with α-MSH induced the phosphorylation of PKA at threonine 197, p38 at threonine 180 and tyrosine 180, and ERK at threonine 202 and tyrosine 204. However, cells co-treated with α-MSH and LRE displayed a considerable decrease in the phosphorylation levels of PKA, p38, and ERK (Figure 5A,B). Furthermore, whether the decreased phosphorylation of ERK, p38, and PKA was time-dependent was investigated. The phosphorylation levels of p38 and ERK were reduced in the α-MSH (200 nM) and LRE (100 µg/mL) co-treated group compared to those in the α-MSH-treated group at 2, 4, and 8 h (Figure 6A–D). In addition, the phosphorylation of PKA exhibited a similar decreasing trend at 4 h and 8 h. Collectively, these results suggest that LRE downregulates the CREB signaling pathway by inhibiting the PKA, p38, and ERK signaling pathways.

### 2.5. Inhibitory Effects of LRE on cAMP-Mediated Melanogenesis in B16F10 Cells

cAMP acts as a secondary messenger and plays a significant role in regulating various functions in benign melanocytes and melanoma cells [82]. cAMP is generated from two distinct sources, transmembrane and soluble adenylyl cyclases, and its degradation is regulated by a family of proteins known as phosphodiesterases [83]. Recent studies have indicated that distinct cAMP signaling pathways promote pigmentation by modifying the expression of melanogenic genes [84]. While the α-MSH/MC1R signaling pathway mediates the crucial regulatory mechanism of melanin production, several cAMP-inducing agents, such as dibutyryl-cAMP (dbcAMP), forskolin, and 3-isobutyl-1-methylxanthine (IBMX), are widely employed in both in vitro and in vivo melanogenic-mimic models [85,86,87]. Thus, we investigated whether LRE, which inhibits α-MSH-induced melanogenesis, exerts an anti-melanogenic effect on other cAMP-mediated melanogenesis pathways. The groups treated with cAMP-inducing agents (dbcAMP, IBMX, and FSK) exhibited a substantial increase in cellular melanin content (Figure 7A); however, the groups co-treated with cAMP inducers and LRE showed a reduction in melanin content compared to the negative control groups. In addition, cAMP inducers increased cellular tyrosinase activity; however, these effects were markedly decreased in cells co-treated with cAMP inducer and LRE in a dose-dependent manner (Figure 7B). These findings suggest the potential of LRE as a natural anti-melanogenic agent that can inhibit cAMP-mediated melanogenesis.

### 2.6. Characterization of LRE via High-Performance Liquid Chromatography-High-Resolution Mass Spectrometry Analysis

Table 1 and Figure 8 present the various organic compounds identified in the LRE using high-performance liquid chromatography-high-resolution mass spectrometry (HPLC-HRMS). The HRMS conditions and chromatographic separation were carried out using an ACQUITY BEH C18 column (1.7 μm, 150 × 2.1 mm) with a 2 μL injection of LRE. Mass spectrometry acquisition was performed in the positive ion mode, covering the *m/z* range of 150 to 1500. Data analysis for all LRE compositions was conducted using Xcalibur software version 4.3. Figure 8 demonstrates the presence of 13 main compositions in the LRE, and retention times (RTs) for analytes are reported in Table 1. Out of these, two are unknown compounds with RTs of 8.03 and 12.78, and their molecular compositions are C_69_H_75_O_7_ and C_83_H_85_O_3_, respectively. The analytes of interest included various amino acid analogs, namely L-alanyl-L-alpha-aspartyl-L-proline, methyl N-acetylhistidinate, L-phenylalanine, Boc-O-methyl-L-threonine, threonyl-α-glutamylleucine, Boc-Lys(Z)-OH, Z-L-Pro-L-Leu-Gly, Methyl (4S)-4-[(2-pyridinylcarbonyl)amino]-L-prolinate, and L-Lysyl-L-leucyl-L-valyl-L-leucyl-L-alanyl-L-serine, all of which were found within the LRE. Notably, regaloside A, a distinctive bioactive component previously reported in another study, was detected in the LRE [88]. These findings indicate that the organic compounds in the LRE possess anti-melanogenic effects in B16F10 cells. Based on these data, we analyzed the effects of L-phenylalanine and regaloside A on cell viability and its anti-melanogenic properties in B16F10 cells. A WST-1-based cytotoxicity assay revealed the non-toxic nature of L-phenylalanine and regaloside A, with concentrations of up to 500 µM and 200 µM, respectively (Figure 9A,C). Subsequently, whether L-phenylalanine and regaloside A downregulate melanin synthesis was investigated. L-phenylalanine and Regaloside A significantly attenuated melanin content compared with the α-MSH-induced negative control group (Figure 9B,D). Our results are consistent with a previous study that suggests L-phenylalanine decreases melanin synthesis by inhibiting the uptake of tyrosine [89]. Taken together, these results suggest that LRE, which contains diverse organic compounds, such as glucosides and amino acids, along with regaloside A, may exhibit synergistic anti-melanogenic effects in B16F10 cells.

## 3. Discussion

Despite numerous studies elucidating the beneficial physiological effects of *L. lancifolium*, including its anti-inflammatory and antioxidant properties, there is a scarcity of research on its physiological interactions within the skin [56,58,90]. Notably, the leaves, roots, and bulbs of *L. lancifolium* have been utilized in medicinal practices in Northeast Asia and are recognized for their amino acids, polysaccharides, saponins, phenylpropanoids, phenolics, and other compounds, suggesting their potential to ameliorate skin conditions [57,58,59,60]. Therefore, this study aimed to investigate the anti-melanogenic effects of *L. lancifolium* on B16F10 cells, anticipating its physiological role in the skin. Water was used as the extraction solvent for this purpose.

Skin pigmentation is a highly conserved defense mechanism in skin tissues against deleterious factors, such as UV radiation [91]. However, excessive melanin production can result in pigmentary disorders, such as hyperpigmentation, melasma, and solar lentigines [92,93]. Considering the impact of facial hyperpigmentation on the quality of life of patients, there is an imperative need for novel cosmetic ingredients derived from natural sources to alleviate hyper-melanogenic conditions [94]. Therefore, we investigated the anti-melanogenic effects of LRE in B16F10 cells. Our findings demonstrated that LRE effectively reduced melanin production in a dose-dependent manner without causing cytotoxicity (Figure 1A,B). In previous studies, representative melanogenic enzymes, including tyrosinase, Tyrp1, and Tyrp2, have been identified as crucial elements that upregulate melanogenesis in melanocytes. Therefore, we examined whether the LRE-induced reduction in melanin production could be attributed to the downregulation of these melanogenic enzymes. Our results revealed that LRE reduced cellular tyrosinase activity as well as the mRNA and protein levels of tyrosinase, Tyrp1, and Tyrp2 in a dose-dependent manner (Figure 2A–D). Furthermore, the melanin-reducing efficacy of LRE (100 µg/mL) notably surpassed that of arbutin (100 µM). This emphasizes the potential of LRE to efficiently exert anti-melanogenic effects by impeding the expression of core melanogenic enzymes in B16F10 cells.

The Intricate process of melanogenesis is governed by an enzymatic cascade, including tyrosinase, which is modulated by transcription factors, such as Mitf and CREB [72]. α-MSH binding to MC1R triggers cAMP production, which induces the phosphorylation of the CREB transcription factor, increasing Mitf expression. Mitf binds to the promoter regions of melanin-producing genes and positively regulates the transcription of tyrosinase, Tyrp1, and Tyrp2 [67,72,95,96]. Therefore, inhibition of cAMP/CREB/Mitf signaling in B16F10 cells has been explored in various studies as a therapeutic strategy for mitigating hyperpigmentation [97,98]. In this study, LRE effectively attenuated CREB/Mitf in a time- and dose-dependent manner both in the presence and absence of α-MSH treatment (Figure 3 and Figure 4C). In particular, co-treatment with LRE and α-MSH exhibited a reduction in the phosphorylation level of CREB at the serine 133 residue, which was elevated by α-MSH treatment. In addition, Mitf protein stability was reduced by LRE treatment (Figure 3B). These findings suggest that LRE inhibits melanogenesis by effectively attenuating the CREB/Mitf signaling pathway.

Multiple kinases have been identified as regulators of CREB, the transcription factor modulating the transcriptional level of Mitf [73]. Several studies have demonstrated that the key kinases, such as PKA, p38, and ERK, are phosphorylated by increased cAMP levels induced by α-MSH, ultimately enhancing the CREB/Mitf pathway [74,75,76,78,79,80]. Therefore, to comprehend the molecular mechanism by which LRE inhibits the phosphorylation of CREB at the serine 133 residue, the upstream kinases of CREB were assessed using Western blotting. Our findings demonstrated that LRE reduced the phosphorylation levels of ERK, p38, and PKA, even when administered as a single treatment. Furthermore, LRE decreased the elevated phosphorylation levels of ERK, p38, and PKA induced by α-MSH treatment (Figure 5A,B). In addition, co-treatment with LRE decreased the phosphorylation levels of ERK, p38, and PKA, which were elevated by α-MSH (200 nM) treatment at 4 h and 8 h (Figure 6A–D). Collectively, these findings suggest that LRE inhibits Mitf expression and melanin synthesis by suppressing the PKA/CREB and mitogen-activated protein kinase (MAPK)/CREB signaling pathways. Subsequently, to confirm whether the anti-melanogenic effect of LRE occurs in melanogenesis models stimulated by cAMP-inducing agents distinct from α-MSH, melanin content and cellular tyrosinase assays were performed. LRE inhibited melanogenesis induced by dbcAMP, IBMX, and forskolin in a dose-dependent manner (Figure 7A,B). These results suggest that LRE inhibits cAMP-mediated melanogenesis through the PKA/CREB and MAPK/CREB signaling pathways. Further in-depth validation of the role of LRE in PKA, ERK, and p38 phosphorylation is required.

Furthermore, HPLC-HRMS analysis revealed the presence of various amino acid analogs, including L-phenylalanine, within the LRE. Regaloside A, a bioactive compound unique to *L. lancifolium*, was also present in the LRE [56,88,99]. L-phenylalanine and regaloside A from LRE exhibited potential anti-melanogenic properties in the melanin content assay; however, additional validation is necessary to assess its reliability regarding its anti-melanogenic effects on B16F10 cells and how it affects skin whitening [89].

In this study, we found that LRE may alleviate hyperpigmentation by decreasing the levels of core melanogenic elements in cultured B16F10 cells. Despite a single treatment, LRE decreased CREB/Mitf signaling, along with the phosphorylation levels of its upstream signaling proteins, such as PKA, ERK, and p38. Consequently, LRE exhibited a stronger anti-melanogenic effect than arbutin regarding melanin synthesis and intracellular tyrosinase activity. The therapeutic effects of LRE on skin whitening and its definitive molecular mechanisms require further evaluation.

In conclusion, this study provided the first evidence that LRE has the potential to be an effective and safe depigmenting agent. We also demonstrated the anti-melanogenic effects of LRE as a novel cosmetic ingredient that overcomes the side effects of conventional whitening treatments.

## 4. Materials and Methods

### 4.1. Cell Culture and Preparation of L. lancifolium Extract

B16F10 murine melanoma cells were procured from ATCC (Manassas, VA, USA). The cells were cultured in Dulbecco’s Modified Eagle’s medium (Gibco, Grand Island, NY, USA) supplemented with 10% fetal bovine serum (Gibco), 1% streptomycin (100 mg/mL) (Gibco), and penicillin (100 U/mL) (Gibco)at 37 °C in a humid environment with 5% CO_2_. The root of *Lilium lancifolium* utilized in this study was obtained from a cultivation site located in the Pyeongchang region (Gangwon-do, Republic of Korea) in May 2022. The extraction method of *L. lancifolium root* was performed according to the previously described method [100]. After three cycles of rinsing with distilled water, the plants were left to air dry at room temperature. Ten grams of dried *L. lancifolium* root were chopped and subsequently finely pulverized into a fine powder using a grinder (SMX-3500GN, Shinil Industrial Co. Ltd., Seoul, Republic of Korea). Following this, the finely ground *L. lancifolium* root powder (10 g) was mixed with 200 mL of hot distilled water (80 °C) and left to steep for 4 h. The mixture was initially filtered using Whatman filter paper No. 1 (Whatman, Maidstone, UK) and subsequently, an additional purification step was carried out by subjecting it to ultrafiltration through a sterile 0.2 µm bottle-top vacuum filter (Corning, Corning, NY, USA). Then, the filtrate was concentrated using a Rotavapor R-100 rotary evaporator (Buchi, Flawil, Switzerland) under vacuum and then lypophilized in a TFD-100 Freeze Dryer (ilShinBioBase Co., Ltd., Yangju-si, Republic of Korea) for 48 h. The lypophilized water extract from *L. lancifolium* root can be obtained at 1.58 g and the 1 g of freeze-dried water extract was diluted with 10 mL PBS at a concentration of 100 mg/mL. The extracted *L. lancifolium* root was aliquoted into 1 mL portions and stored at −20 °C until use.

### 4.2. HPLC-HRMS Analysis

HPLC-HRMS analysis was conducted following a previously described method [100]. The chemical profiles of LRE were examined using a Thermo Ultimate-3000 UPLC system (Thermo Fisher Scientific, Waltham, MA, USA) coupled with Thermo LTQ-Orbitrap XL (Thermo Fisher Scientific) and executed utilizing an ACQUITY BEH C18 column (1.7 µm, 150 mm × 2.1 mm). The gradient conditions were established in the following sequence: 0 min (5% B), 0–5 min (5% B), 5–20 min (70% B), and 20–27 min (100% B). The flow rate was set to 0.4 mL/min, with an injection volume of 2 µL. The scan range was *m/z* 150–1500 and MS experiments were performed in positive ion mode under the following conditions: FWHM resolution, 60,000; spray voltage, 4.0 kV; capillary voltage 35 V; and capillary temperature 300 °C. The data analysis was carried out utilizing the Xcalibur software (Version 4.3, Thermo Finnigan, San Jose, CA, USA). Regaloside A (MedChemExpress, Monmouth Junction, NJ, USA), known to be uniquely present in *L. lancifolium*, was utilized for quality assessment of its extract through HPLC-HRMS analysis [56].

### 4.3. Cell Viability Assay

To analyze the effect of LRE on cell viability, B16F10 cells treated with LRE were assessed using the WST-1 assay. Briefly, B16F10 cells (3 × 10^3^ cells/well) were initially seeded within a 96-well plate and maintained at 37 °C for 24 h. Subsequently, they were treated with various concentrations of LRE (0–1000 µg/mL) and maintained for 24 h and 48 h. After incubation, a solution of EZ-Cytox (100 µL/well) (DoGenBio, Seoul, Republic of Korea) was added to each well and the mixture was incubated for 30 min at 37 °C. Cell viability was detected by measuring absorbance at 450 nm using a SynergyTM HTX Multi-Mode Microplate Reader (Bioteck, Winooski, VT, USA).

### 4.4. Measurement of Intracellular Melanin Content

Measurement of melanin content was conducted following a previously described method with a slight modification [101]. B16F10 cells (0.5 × 10^5^ cells/mL) were seeded in a 60 mm plate and incubated at 37 °C for 24 h. Afterward, the cells were treated with or without 200 nM α-MSH (Sigma-Aldrich, St. Louis, MO, USA), and then subjected to treatment with various concentrations of LRE (0–1000 µg/mL), regaloside A (0–200 µM), or L-phenylalanine (0–500 µM) (DaejungChemicals, Siheung-si, Republic of Korea). Arbutin (Sigma-Aldrich) was employed as a positive control, while α-MSH (Sigma-Aldrich), dbcAMP (MedChemExpress), IBMX (Sigma-Aldrich), and forskolin (Sigma-Aldrich) were used as negative controls. After 48 h of incubation, the medium was carefully removed, and the cells were washed twice with PBS. Cell pellets were photographed and dissolved in 1 N NaOH lysis buffer at 100 °C for 30 min. The dissolved melanin content was measured at 450 nm using a microplate reader.

### 4.5. Measurement of Tyrosinase Activity Assay

Intracellular tyrosinase activity was measured following a previously described method with a slight modification [41]. B16F10 cells (0.5 × 10^5^ cells/mL) were seeded in a 60 mm plate and cultured for 24 h. Subsequently, the cells were stimulated with or without 200 nM α-MSH, and then treated with various concentrations of LRE (0–1000 µg/mL), regaloside A (0–200 µM), or L-phenylalanine (0–500 µM). After 48 h, the cells were washed twice with PBS and then lysed with RIPA buffer (containing 50 mM pH 7.4 Tris hydrochloride, 150 mM sodium chloride, 1% NP-40, 0.5% sodium deoxy cholate, 0.1% sodium dodecyl sulfate) for 1 h. The lysates were then centrifuged at 10,000× *g* for 30 min. The dissolved supernatant was harvested, and total protein content was quantified using Pierce^TM^ BCA Protein Assay Kit (Thermo Fisher Scientific, Rockford, IL, USA) following the manufacturer’s instructions. Equal amounts of protein were mixed with 20 µL of 10 mM L-DOPA and incubated at 37 °C for 1 h. Absorbance was measured at a wavelength of 490 nm using a microplate reader.

### 4.6. PCR and Quantitative Real-Time PCR

B16F10 cells were cultured in 100 mm dishes. After incubation overnight, the cells were treated with different concentrations of LRE. Total RNA was executed using the RiboEx reagent (Geneall Biotechnology, Seoul, Republic of Korea) as per the manufacturer’s instructions. Synthesis of complementary DNA was performed using 1 µg of total RNA, oligo dT primers, 0.1 M DTT, 2.5 mM dNTPs, 5X First-Strand Buffer, and M-MLV reverse transcriptase (Thermo Fisher Scientific, Waltham, MA, USA). Glyceraldehyde-3-phosphate dehydrogenase (*Gapdh*) was used as a reference for the gene expression level normalization. The primer sequence for the specific gene analyzed is provided in Table 2.

### 4.7. Western Blot Analysis

B16F10 cells (0.5 × 10^5^ cells/mL) were seeded in a 60 mm plate and cultured for 24 h. The cells treated with indicated concentrations of LRE and α-MSH were lysed in Radioimmunoprecipitation (RIPA) assay buffer. The total protein content was measured by Pierce BCA Protein Assay Kit (Thermo Fisher Scientific). Proteins were loaded via 8% or 10% sodium dodecyl sulfate polyacrylamide gel electrophoresis (SDS-PAGE) and transferred to a nitrocellulose membrane. The membrane was blocked with 5% skim milk and incubated overnight at 4 °C with primary antibodies. The blots were incubated with horseradish peroxidase-conjugated anti-mouse IgG (#7076S; Cell Signaling Technology; CST; Danvers, MA, USA) or anti-rabbit IgG (#7074S; CST) secondary antibodies. Proteins were detected with ECL reagent (Bio-Rad) and visualized by the ChemiDoc Touch Imaging System (Bio-Rad, Hercules, CA, USA). The protein samples 20 µg protein samples were examined via Western blotting with the corresponding antibodies. All antibodies were purchased from Santa Cruz (Dallas, TX, USA) or CST. The following antibodies are listed in Table 3.

### 4.8. Statistical Analysis

Data was analyzed by one-way analysis of variance (ANOVA) to determine the statistical significance of variations among the treatment groups. Subsequently, in instances where statistically significant treatment effects were detected, Tukey’s test was used for comparisons between the means of multiple groups. The data are expressed as the mean ± standard deviation (SD). Statistical significance was attributed to differences with *p* < 0.05.

## Figures and Tables

**Figure 1 plants-12-03666-f001:**
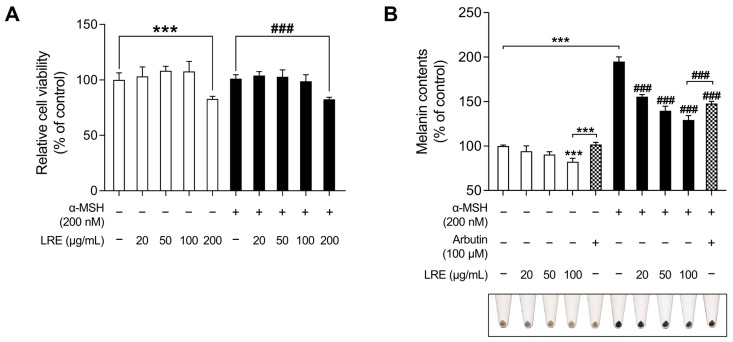
Effects of *Lilium lancifolium* (*L. lancifolium)* root extract on cell viability and melanin production in B16F10 cells. (**A**) B16F10 cells were seeded in 96-well plates (2 × 10^3^ cells/well) and incubated for 24 h. The cells were treated with the indicated concentrations of LRE and α-MSH for 48 h. Cell viability of B16F10 was measured via a WST-1 assay. (**B**) The cells were seeded in 60 mm dishes (1 × 10^5^ cells) and incubated for 24 h. The cells were treated with indicated concentrations of LRE, α-MSH, and arbutin for 48 h. Arbutin was utilized as the positive control. Intracellular melanin content was measured following stimulation with or without α-MSH and subsequent treatment with LRE. The results are presented as the mean ± SD of three independent experiments and were analyzed using a one-way analysis of variance followed by Tukey’s test. LRE, *Lilium lancifolium* root extract; WST-1, water-soluble tetrazolium salt-1; α-MSH, alpha-melanocyte-stimulating hormone. ^###,^ *** *p* < 0.001.

**Figure 2 plants-12-03666-f002:**
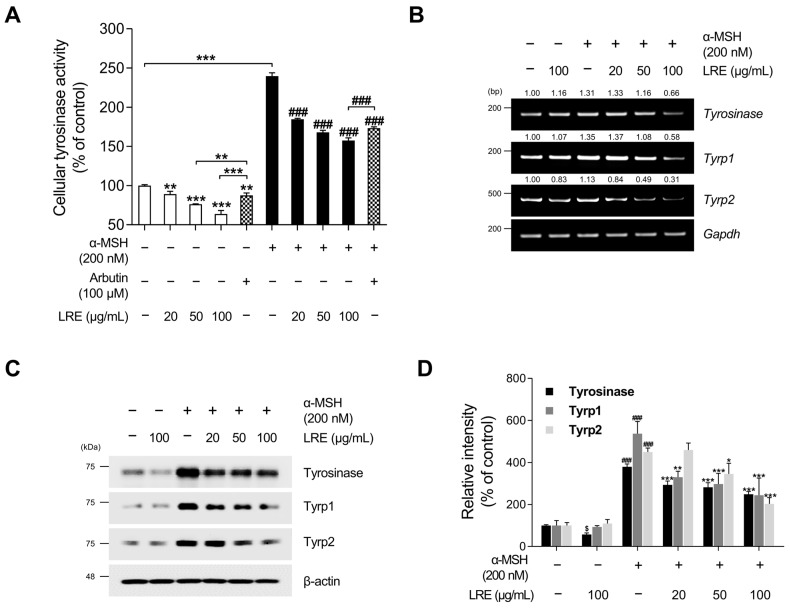
Effects of *L. lancifolium* root extract on cellular tyrosinase activity and melanogenic enzyme expression. (**A**) B16F10 cells were seeded in 60 mm dishes (1 × 10^5^ cells) and incubated for 24 h. The cells were treated with the indicated concentrations of LRE, α-MSH, and arbutin for 48 h. Intracellular tyrosinase activity was assessed following treatment of B16F10 cells with LRE and stimulation with or without α-MSH. (**B**) B16F10 cells were seeded in 60 mm dishes (1 × 10^5^ cells) and incubated for 24 h. The cells were then treated with the indicated concentrations of LRE and α-MSH for 24 h. mRNA levels of tyrosinase-related genes (*tyrosinase*, *Tyrp1*, and *Tyrp2*) were detected via RT-PCR, with *GAPDH* serving as a loading control. (**C**) B16F10 cells were incubated with the indicated concentrations of LRE and α-MSH for 48 h. Protein levels of tyrosinase-related enzymes (tyrosinase, Tyrp1, and Tyrp2) were analyzed via Western blotting, with β-actin serving as a loading control. (**D**) Quantitation of protein level was conducted using ImageJ software version 1.53t. The results are presented as the mean ± SD of three independent experiments and were analyzed using a one-way analysis of variance followed by Tukey’s test. RT-PCR, reverse-transcription polymerase chain reaction; mRNA, messenger RNA. ^$,^ * *p* < 0.05; ** *p* < 0.01; ^###,^ *** *p* < 0.001.

**Figure 3 plants-12-03666-f003:**
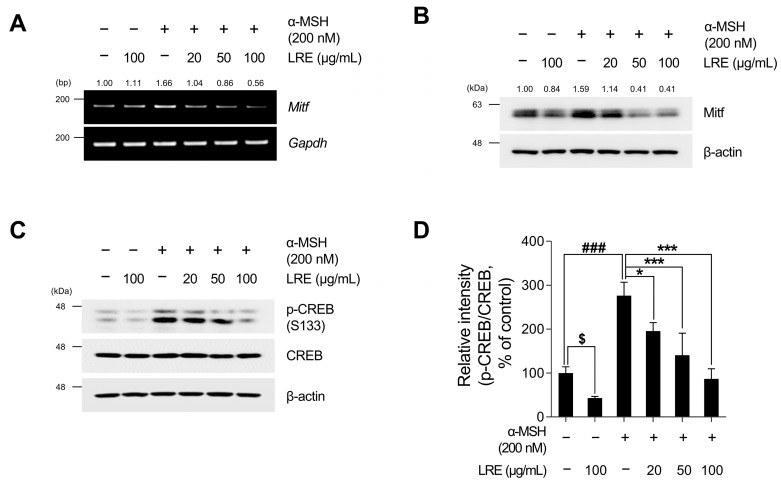
Effects of *L. lancifolium* root extract on the CREB/Mitf signaling pathway. (**A**) B16F10 cells were seeded in 60 mm dishes (2 × 10^5^ cells) and incubated for 24 h. The cells were treated with the indicated concentrations of LRE and α-MSH for 24 h. The mRNA levels of *Mitf* were detected via RT-PCR, with *GAPDH* serving as a loading control. (**B**) The cells were incubated with the indicated concentrations of LRE and α-MSH for 24 h. The Mitf protein levels were analyzed via Western blotting, while β-actin served as a loading control. (**C**) The cells were incubated with the indicated concentrations of LRE and α-MSH for 12 h. The protein levels and phosphorylation of CREB were analyzed via Western blotting, while β-actin served as a loading control. (**D**) Quantitation of protein levels was conducted using ImageJ software version 1.53t. The results are presented as the mean ± SD of three independent experiments and were analyzed using a one-way analysis of variance followed by Tukey’s test. CREB, cyclic adenosine monophosphate response element-binding protein; Mitf, microphthalmia-associated transcription factor. ^$,^ * *p* < 0.05; ^###,^ *** *p* < 0.001.

**Figure 4 plants-12-03666-f004:**
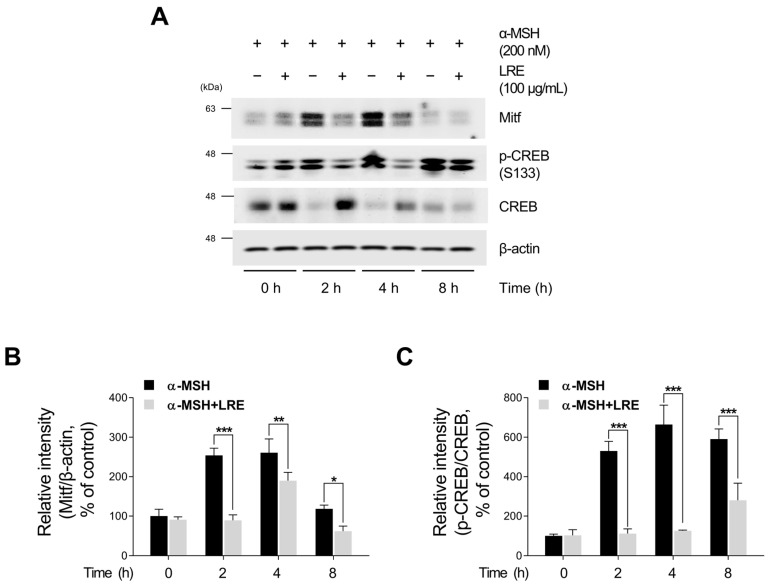
Effects of *L. lancifolium* root extract on the CREB/Mitf signaling pathway, depending on treatment duration. (**A**) Mitf, p-CREB, and CREB protein levels were examined at different time points after co-treatment with LRE (100 µg/mL) and α-MSH (200 nM). The protein levels were determined via Western blotting, while β-actin served as a loading control. (**B**,**C**) Quantitation of protein levels was conducted using ImageJ software version 1.53t. The results are presented as the mean ± SD of three independent experiments and were analyzed using a one-way analysis of variance followed by Tukey’s test. * *p* < 0.05; ** *p* < 0.01; *** *p* < 0.001.

**Figure 5 plants-12-03666-f005:**
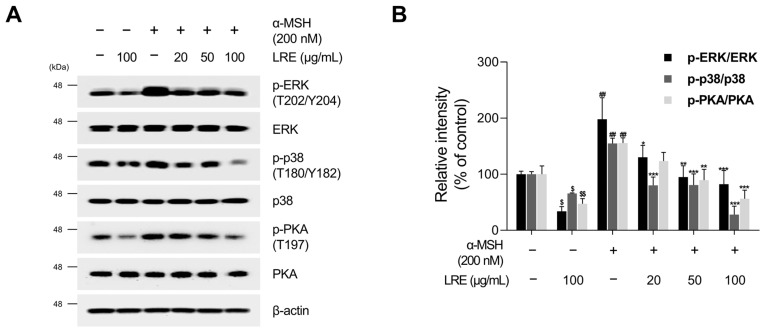
Effects of *L. lancifolium* root extract on the PKA/CREB and MAPK/CREB signaling pathways. (**A**) B16F10 cells were seeded in 60 mm dishes (2 × 10^5^ cells) and incubated for 24 h. The cells were treated with the indicated concentrations of LRE and α-MSH for 12 h. The protein levels of CREB upstream signaling were assessed via Western blotting, with β-actin serving as a loading control. (**B**) Protein levels were quantified using ImageJ software version 1.53t. The results are presented as the mean ± SD of three independent experiments and were analyzed using a one-way analysis of variance followed by Tukey’s test. PKA, protein kinase A; MAPK, mitogen-activated protein kinase. *^, $^ *p* < 0.05; **^, ##, $$^ *p* < 0.01; *** *p* < 0.001.

**Figure 6 plants-12-03666-f006:**
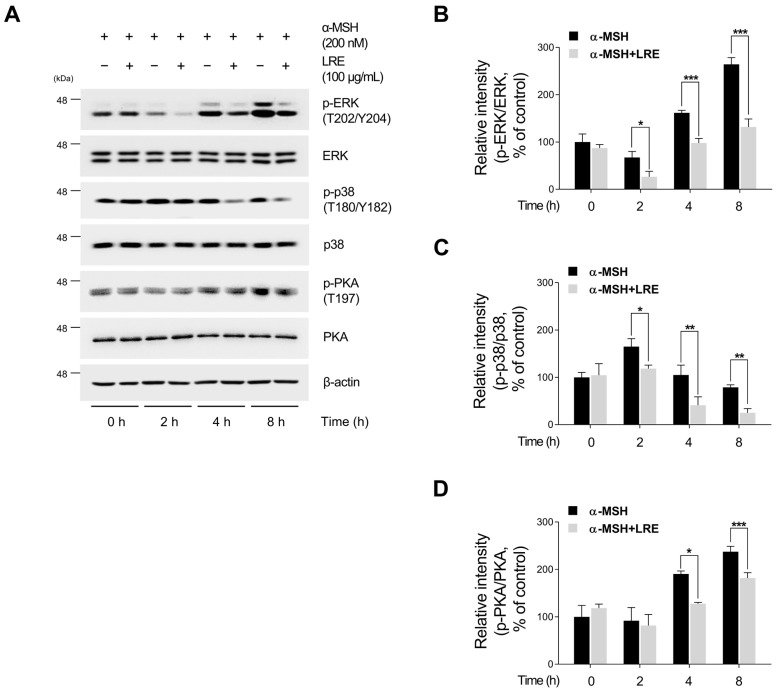
Effects of *L. lancifolium* root extract on the PKA/CREB and MAPK/CREB signaling pathways depending on treatment duration. (**A**) B16F10 cells were seeded in 60 mm dishes (2 × 10^5^ cells) and incubated for 24 h. The cells were treated with the indicated concentrations of LRE and α-MSH for 2, 4, and 8 h. The protein levels of CREB upstream signaling were assessed via Western blotting, with β-actin serving as a loading control. (**B**–**D**) Protein levels were quantified using ImageJ software version 1.53t. The results are presented as the mean ± SD of three independent experiments and were analyzed using a one-way analysis of variance followed by Tukey’s test. ERK, extracellular signal-related kinase. * *p* < 0.05; ** *p* < 0.01; *** *p* < 0.001.

**Figure 7 plants-12-03666-f007:**
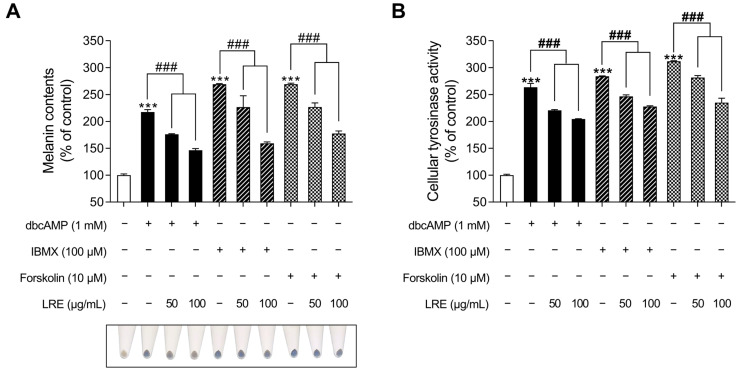
Effects of *L. lancifolium* root extract on cAMP-mediated melanogenesis. (**A**) B16F10 cells were seeded in 60 mm dishes (1 × 10^5^ cells) and incubated for 24 h. The cells were treated with indicated concentrations of LRE and cAMP inducer (dbcAMP, IBMX, and FSK) for 48 h. Intracellular melanin contents were determined following stimulation with or without cAMP inducer and treatment with LRE. (**B**) The cells were seeded in 60 mm dishes (1 × 10^5^ cells) and incubated for 24 h. The cells were treated with the indicated concentrations of LRE and cAMP inducer for 48 h. Intracellular tyrosinase activity was determined following treatment of B16F10 cells with LRE and stimulation with or without cAMP inducer. The results are presented as the mean ± SD of three independent experiments and were analyzed using a one-way analysis of variance followed by Tukey’s test. dbcAMP, dibutyryl-cAMP; IBMX, 3-isobutyl-1-methylxanthine; FSK, forskolin; cAMP, cyclic adenosine monophosphate. ***^, ###^ *p* < 0.001.

**Figure 8 plants-12-03666-f008:**
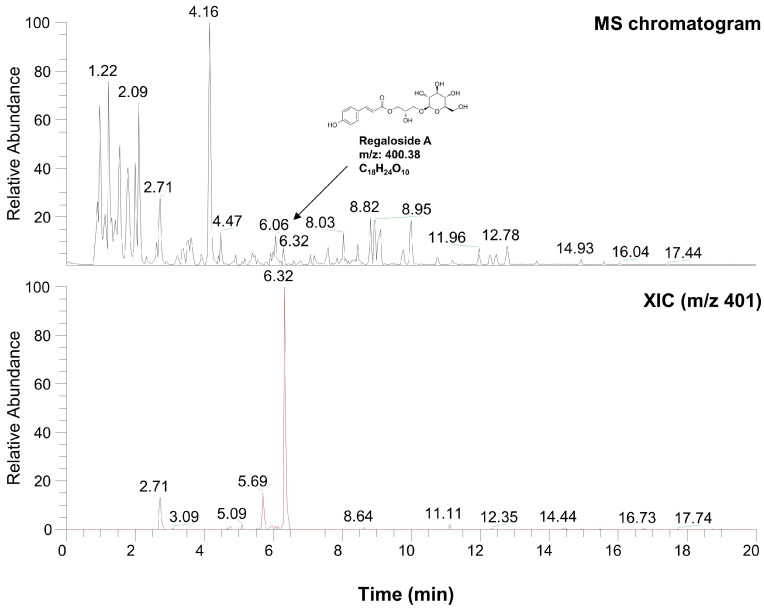
Characterization of compounds from *L. lancifolium* extract using high-performance liquid chromatography-high-resolution mass spectrometry analysis.

**Figure 9 plants-12-03666-f009:**
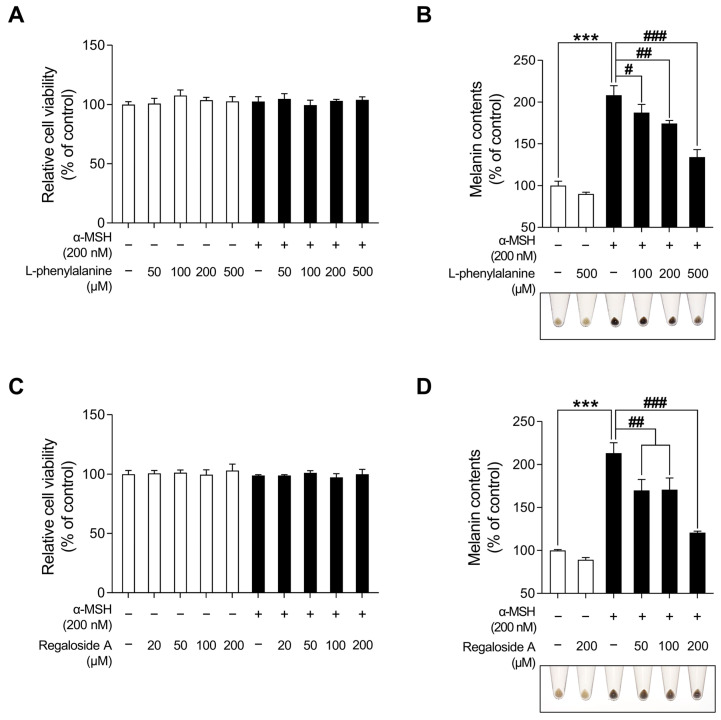
Effects of L-phenylalanine and regaloside A on cell viability and melanin production in B16F10 cells. (**A**,**C**) B16F10 cells were seeded in 96-well plates (2 × 10^3^ cells/well) and incubated for 24 h. The cells were treated with the indicated concentrations of L-phenylalanine or reglaoside A, with or without α-MSH for 48 h. Cell viability of B16F10 was measured via the WST-1 assay. (**B**,**D**) The cells were seeded in 60 mm dishes (1 × 10^5^ cells) and incubated for 24 h. The cells were treated with the indicated concentrations of L-phenylalanine or regaloside A, with or without α-MSH for 48 h. Intracellular melanin contents were determined following stimulation with or without α-MSH and treatment with L-phenylalanine or regaloside A. The results are presented as the mean ± SD of three independent experiments and were analyzed using a one-way analysis of variance analysis followed by Tukey’s test. ^#^
*p* < 0.05; ^##^ *p* < 0.01; ***^, ###^ *p* < 0.001.

**Table 1 plants-12-03666-t001:** High-resolution mass spectrometry data of identified molecules in *L. lancifolium* extract.

Number	Name	Formula	M.W.	RT (min)
1	L-alanyl-L-alpha-aspartyl-L-proline	C_12_H_19_O_6_N_3_	301.296	1.22
2	Methyl N-acetylhistidinate	C_9_H_13_O_3_N_3_	211.218	1.68
3	L-phenylalanine	C_9_H_11_NO_2_	165.189	2.09
4	Threonyl-α-glutamylleucine	C_15_H_27_O_7_N_3_	361.391	2.71
5	Boc-O-methyl-L-threonine	C_10_H_19_NO_5_	234.133	4.16
6	Hopantenic acid glucoside	C_16_H_29_O_10_N	395.402	4.47
7	Methyl (4S)-4-[(2-pyridinylcarbonyl)amino]-L-prolinate	C_12_H_15_O_3_N_3_	250.118	6.06
8	Regaloside A	C_18_H_24_O_10_	400.377	6.32
9	Boc-Lys(Z)-OH	C_19_H_28_O_6_N_2_	380.435	8.82
10	Z-L-Pro-L-Leu-Gly	C_21_H_29_O_6_N_3_	419.471	8.95
11	L-Lysyl-L-leucyl-L-valyl-L-leucyl-L-alanyl-L-serine	C_29_H_55_O_8_N_7_	629.789	11.96

**Table 2 plants-12-03666-t002:** List of sequences used for PCR and qRT-PCR.

Target mRNA	Sequences of Primer	Amplicons (bp)	Annealing Temperature (°C)
*Gapdh*	F: 5′-CATCACTGCCACCCAGAAGACTG-3′	153	60
	R: 5′-ATGCCAGTGAGCTTCCCGTTCAG-3′		
*Mitf*	F: 5′-AGAAGCTGGAGCATGCGAACC-3′	168	60
	R: 5′-GTTCCTGGCTGCAGTTCTCAAG-3′		
*Tyrosinase*	F: 5′-AGTCGTATCTGGCCATGGCTTCTTG-3′	169	60
	R: 5′-GCAAGCTGTGGTAGTCGTCTTTGTC-3′		
*Tyrp1*	F: 5′-CTGCGATGTCTGCACTGATGACTTG-3′	171	60
	R: 5′-TTTCTCCTGATTGGTCCACCCTCAG-3′		
*Tyrp2*	F: 5′-GCTTGGATGACTACAACCGCCG-3′	451	60
	F: 5′-GGTGGGAAGAAGGGGACCATGT-3′		

**Table 3 plants-12-03666-t003:** List of primary antibodies for Western blot analyses.

Antigen	Host	Clonality (Species Reactivity)	Dilution	Manufacturer (Cat. Number)	References
Mitf	Mouse	Monoclonal (mouse, rat, human)	1:200	Santa Cruz (#sc-56725)	[102,103]
Tyrosinase	Mouse	Monoclonal (mouse, rat, human)	1:200	Santa Cruz (#sc-20035)	[102,104]
Tyrp1	Mouse	Monoclonal (mouse, rat, human)	1:200	Santa Cruz (#sc-166857)	[105,106]
Tyrp2	Mouse	Monoclonal (mouse, rat, human)	1:200	Santa Cruz (#sc-74439)	[107,108]
β-actin	Mouse	Monoclonal (mouse, rat, human)	1:1000	Santa Cruz (#sc-47778)	[109,110]
PKA	Rabbit	Polyclonal (mouse, rat, human)	1:1000	CST (#4782S)	[111,112]
Phospho-PKA (Thr198)	Rabbit	Polyclonal (mouse, rat, human)	1:1000	CST (#4781S)	[112,113]
CREB	Rabbit	Monoclonal (mouse, rat, human···)	1:1000	CST(#9197S)	[31,114]
Phospho-CREB (Ser133)	Rabbit	Monoclonal (mouse, rat, human···)	1:1000	CST (#9198S)	[31,109]
ERK	Rabbit	Polyclonal (mouse, rat, human···)	1:1000	CST (#9102S)	[106,115]
Phospho-ERK (Thr202/Tyr204)	Rabbit	Polyclonal (mouse, rat, human···)	1:1000	CST (#9101S)	[106,115]
p38	Rabbit	Polyclonal (mouse, rat, human···)	1:1000	CST (#9212S)	[116,117]
Phospho-p38 (Thr180/Tyr182)	Rabbit	Polyclonal (mouse, rat, human···)	1:1000	CST (#9211S)	[111,116]

## Data Availability

Not applicable.

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
