# Peer review of "Anti-Melanogenic Effects of Lilium lancifolium Root Extract via Downregulation of PKA/CREB and MAPK/CREB Signaling Pathways in B16F10 Cells"

_plants, 2023, doi:10.3390/plants12213666_

Round 1

Reviewer 1 Report

This paper described the effect of Lilium root extract on melanin synthesis.

Authors performed several types of experiments, and the process of study is acceptable. 

They predicted that the regaloside is the main anti-melanogenic compound in root extract.

This paper is interesting, however, I found that there are some problems in this manuscript.

Author should add line number during review process.

Introduction is well-written and contains abundant information and backgrounds.

Results

Figures resolution is very low. Authors should replace the better figures. 

According to Fig. 2b (RT-PCR), I could not find the duration time of LTE treatment.

Also, I could not find quantitative RT-PCR data. Authors should add the data.

Methods

4.1 according to cell culture of B16F10, there is a duplication of sentence (media, antibiotics…).

Reviewer 2 Report

INTRODUCTION.

 The hydroquinone paragraph should be corrected as it contains many factual inaccuracies.

RESULTS

 The SDs of the graphs in Figures 3-6 are too small; Western blotting images should be included for all three cases.

 MS chromatography data for the molecular weight of 6.06 min in Figure 8 should be included.

Methods.

All antibodies should be listed with their serial numbers and whether they are rabbit or mouse antibodies. Also, indicate whether all antibodies correctly recognize the protein of interest with references.

INTRODUCTION.

 The hydroquinone paragraph should be corrected as it contains many factual inaccuracies.

RESULTS

 The SDs of the graphs in Figures 3-6 are too small; Western blotting images should be included for all three cases.

 MS chromatography data for the molecular weight of 6.06 min in Figure 8 should be included.

Methods.

All antibodies should be listed with their serial numbers and whether they are rabbit or mouse antibodies. Also, indicate whether all antibodies correctly recognize the protein of interest with references.

Reviewer 3 Report

1.     Please indicate the greatest innovation of the research conducted and the results obtained

2.     What was the final concentration of the extract? There is information that 10 grams of the raw material was poured with 200 ml of water and subjected to 3 cycles. It is known that you will not recover 200 ml each time. So was the extract later concentrated to some volume? If so, to which one?

3.     Since in the methodology part the Authors describe the use of the HPLC-HRMS method, which was used for qualitative determination of the content of ingredients in the extract (chapter 4.2), information about the analysis results should also be included in the beginning of results part. The section of results begins with cell research. Information about HPLC analysis is included only in section 2.6, so the order of results and methodology is not consistent.

4.     MS analysis allows the identification of the full composition of active compounds, why has this not been done? The regaloside A peak in the chromatogram is not the highest, so why was this compound identified? There are other ingredients with higher concentrations in the extract and shouldn't the Authors also check their individual effects on cells?

5.     The quality of all figures should be improved.

Round 2

Reviewer 1 Report

The paper was appropriately modified. 

Reviewer 2 Report

It is better than the first edition, so there is no problem to publish it.

It is better than the first edition, so there is no problem to publish it.

Reviewer 3 Report

Accept in present form